# A Longitudinal Assessment of Substance Use Treatment during the COVID-19 Pandemic Using Staff and Service Data

Molly Carlyle [ID], Grace Newland [ID], Leith Morris, Rhiannon Ellem, Calvert Tisdale, Catherine A. Quinn and Leanne Hides *

Lives Lived Well Research Group, National Centre for Youth Substance Use Research, School of Psychology, Faculty of Health and Behavioural Sciences, University of Queensland, Brisbane 4072, Australia
* Correspondence: l.hides@uq.edu.au

**Abstract:** Introduction: Alcohol and other drug (AOD) treatment services were required to rapidly adapt delivery of care in response to the coronavirus (COVID-19) pandemic. This study examined longitudinal changes in the delivery of AOD counselling in Australia over 21 months (October 2019–July 2021) before and throughout the pandemic, using both staff self-report and service data. Methods: Treatment staff from a large AOD service in Queensland, Australia provided self-report data on time spent delivering counselling via face-to-face, outreach (home visits), telephone, and virtual (video) formats. Two waves of online questionnaires were collected, with staff reporting on their time before the pandemic (retrospectively for October 2019–February 2020); during the first lockdown period (retrospectively for March–May 2020); when restrictions were initially eased (June–September 2020); and one year later (July 2021). Service records of the number of counselling episodes conducted by each treatment modality were extracted between October 2019 and July 2021, and analysed by month. Results: Staff (*n* = 117) and service records indicated an increase in telephone-delivered AOD counselling during the first lockdown, alongside an increase in total counselling records. Telephone-delivered counselling was still significantly higher one year later. Face-to-face counselling declined after the onset of the pandemic, but increased quickly when restrictions were eased. Outreach counselling decreased during the first lockdown. Virtual counselling remained negligible throughout. Conclusion: AOD treatment services quickly utilised telephone counselling options at the start of the pandemic, and demonstrated continued utilisation of this method one year later. Increased virtual (video) counselling was not observed and may be due to limited infrastructure, staff training, and clients lacking Internet connectivity or technology required.

**Keywords:** COVID-19; substance use; alcohol and other drugs; telehealth; counselling

## 1. Introduction

The coronavirus (COVID-19) pandemic exacerbated alcohol and other drug (AOD) use among people in treatment for substance use disorders [1,2]. In Australia, stay-at-home orders were enforced on 23 March 2020, limiting access to AOD counselling and treatment services. With limited opportunity for face-to-face contact, treatment services were required to adapt quickly in order to protect staff and continue delivering treatment to people with increasing substance use problems, while adhering to social distancing requirements.

Many AOD services reported an increase in the uptake of telephone and virtual (video) counselling [3–5] following early recommendations to utilise alternative treatment delivery methods to ensure continuity of care [6–8]. AOD staff expressed a willingness to adopt changes such as telehealth delivery of care through COVID-19 [9], and clients expressed openness to telehealth-delivered treatment—particularly for virtual (video) treatments [3,10]. However, little research to date has assessed the long-term impacts of COVID-19 on service delivery from both AOD staff perspectives and objective treatment record data.

This study attempts to address this gap by assessing changes in the delivery of AOD counselling over 21 months, including before and throughout the COVID-19 pandemic (October 2019–July 2021). Two sources of data were collected from a large treatment service in Australia: (i) self-report data from two staff surveys during and after the enforcement of pandemic-related restrictions; and (ii) treatment record data extracted from the service's electronic management system. Modality of counselling was compared before the pandemic (October 2019–February 2020); during the first and strictest lockdown (March–May 2020); when restrictions initially eased (June–September 2020); and one year later (July 2021), to assess the uptake and continued use of telehealth services in AOD treatment as a result of the COVID-19 pandemic.

## 2. Materials and Methods

### 2.1. Recruitment and Participants

All clinical staff in a large substance use treatment service located in Queensland and New South Wales, Australia (approx. 204 services; 16,000 clients/year) were invited to complete an annual survey to obtain their views about relevant organisational issues. Inclusion criterion for participation in the current study was that staff had direct client contact as part of their role (either through provision or supervision of counselling). Eligible staff were electronically provided with information about the study, including that their participation was confidential and voluntary, and a consent form to indicate whether they provided consent to participate in the study or not. Information on staff consent and study participation remained with independent research staff only and was not accessible by members of the organisation.

Eligible and consenting staff who completed both survey occasions were included in this study (staff demographics, Supplementary Materials Table S1). The two data collection points were July–September 2020 ($n = 158$), and June–July 2021 ($n = 180$). A total of 221 clinical staff completed the surveys; 117 (53%) completed both occasions, while 104 (47%) completed one occasion. Treatment record data for counselling episodes were extracted from the service's electronic management system for the period of October 2019–June 2021. Ethical approval was obtained from the University of Queensland Office of Research Ethics (IDs 2019002308 and 2020002322).

### 2.2. Measures

The organisational survey asked staff to report the proportion of their time spent delivering counselling: face-to-face in the office; face-to-face via outreach (e.g., home visits); via telephone; and virtually (video call). In the first survey, staff were asked to retrospectively report these proportions (i) for the time period before lockdown (before March 2020); (ii) for the time period during the first lockdown (March–May 2020); and (iii) at present (July–September 2020). The second survey asked staff to report on the proportion of counselling delivered by each modality at the time of the survey only (June 2021).

Staff entered routine counselling record data into the service's electronic management system after each appointment, which included counselling modality (face-to-face, outreach, telephone, virtual). Data were extracted for the October 2019–July 2021 time period on 20 August 2021.

### 2.3. Statistical Analysis

Data were analysed using SPSS Version 25. Cases were retained pairwise. Staff survey responses were analysed using a series of one-way, repeated measures ANOVAs (>2 response options) or *t*-tests (2 response options) for 117 staff that had completed the two survey waves. *t*-tests were also used to assess demographic differences between staff that had completed one ($n = 104$) compared with two ($n = 117$) surveys. The Huynh–Felt correction was used where sphericity was violated. Significant effects were examined using post hoc, Bonferroni-corrected pairwise comparisons.

Treatment record data were quantified and presented descriptively on a monthly basis. Significant differences in treatment records were tested using mixed measure ANOVAs on the same four time periods as staff self-report data. Treatment episode counts for each month were stardardised to z-scores for each delivery type to account for fluctuation in the total number of records that were not attributable to the pandemic.

## 3. Results

### 3.1. Treatment Staff Self-Report Data

The 117 staff that completed both organisational surveys were aged 40.72 years on average (SD = 11.56). The staff reported having worked in the AOD sector for 70.58 months (SD = 73.16), and in this specific service for 44 months (SD = 49.07). The professional training backgrounds of staff included social work ($n = 28$, 24%), counselling ($n = 28$, 24%), AOD work ($n = 23$, 20%), psychology ($n = 9$, 8%), nursing (3%), or other ($n = 25$, 21%). Of the clinical services staff provided, 56 (48%) reported that they were AOD-specific, 38 (33%) were combined AOD and mental health, and 6 (5%) were mental-health-specific. Twenty-three (20%) were in supervisory or managerial roles for clinical services.

While 117 (53%) staff completed both waves of questionnaires, 104 (47%) staff only completed one of the two waves. When comparing differences between these staff, there were no significant differences in the age (t = 0.65, $p = 0.514$) or months of experience in the AOD sector (t = 1.25, $p = 0.214$) between staff who completed one compared with two surveys. Staff who completed only one survey had been working in the service for significantly fewer months (M = 24.19, SD = 25.07) than staff who completed both surveys (t = 3.73, $p < 0.001$).

Repeated measures ANOVAs demonstrated significant changes in the proportion of time that staff spent delivering face-to-face, outreach, and telephone counselling throughout the pandemic period (see Table S2 for ANOVA outcomes). An inverse relationship was observed, whereby face-to-face counselling significantly decreased during the first lockdown period, and telephone counselling increased correspondingly (Figure 1). These changes reverted to pre-pandemic levels once restrictions were eased. There were no significant changes observed in the use of virtual counselling throughout the reporting period.

Pairwise comparisons revealed that the proportion of time spent delivering face-to-face counselling in the office had significantly decreased from before the pandemic (October 2019–February 2020) in comparison to all other timepoints: during the first lockdown (March–May 2020) (MD = 63.36, 95% CI [52.30, 74.42], $p < 0.001$); after restrictions were eased (July–September 2020) (MD = 17.59, 95% CI [6.56, 28.61], $p < 0.001$); and one year later (June 2021) (MD = 12.94, 95% CI [1.66, 24.22], $p = 0.018$). Face-to-face delivery significantly increased between the first lockdown and when restrictions were eased (MD = 45.77, 95% CI [35.91, 55.64], $p < 0.001$). There was no significant change between July–September 2020 and June 2021 (MD = 4.64, 95% CI [8.19, 17.47], $p > 0.999$).

The proportion of time delivering telephone counselling was significantly higher than before the pandemic for all other timepoints: during the first lockdown (March–May 2020) (MD = 77.10, 95% CI [69.70, 84.50], $p < 0.001$); when restrictions eased (July–September 2020) (MD = 19.90, 95% CI [11.78, 28.02], $p < 0.001$); and June 2021 (MD = 9.90, 95% CI [1.136, 18.67], $p = 0.018$). The use of telephone counselling was significantly lower than during lockdown by July–September 2020 (MD = 52.20, 95% CI [47.59, 66.80], $p < 0.001$) and remained significantly lower in June 2021 (MD = 67.20, 95% CI [58.89, 76.31], $p < 0.001$). There was a smaller but significant decline in the use of telephone counselling from July–September 2020 to June 2021 (MD = 10.00, 95% CI [0.51, 19.49], $p = 0.033$).

The proportion of time delivering outreach counselling significantly declined from pre-pandemic to the first lockdown (MD = 17.35, 95% CI [7.90, 26.81], $p < 0.001$). The use of outreach counselling once restrictions had eased (July–September 2020) was significantly higher than during lockdown (March–May 2020) (MD = 13.59, 95% CI [6.11, 21.08], $p < 0.001$). The proportion of outreach counselling was maintained at pre-pandemic levels

in both July–September 2020 (when restrictions were eased) and June 2021 (one year later) (MD = 3.76, 95% CI [-1.76, 9.29], *p* = 0.413; MD = 2.34, 95% CI [-5.92, 10.59], *p* > 0.999).

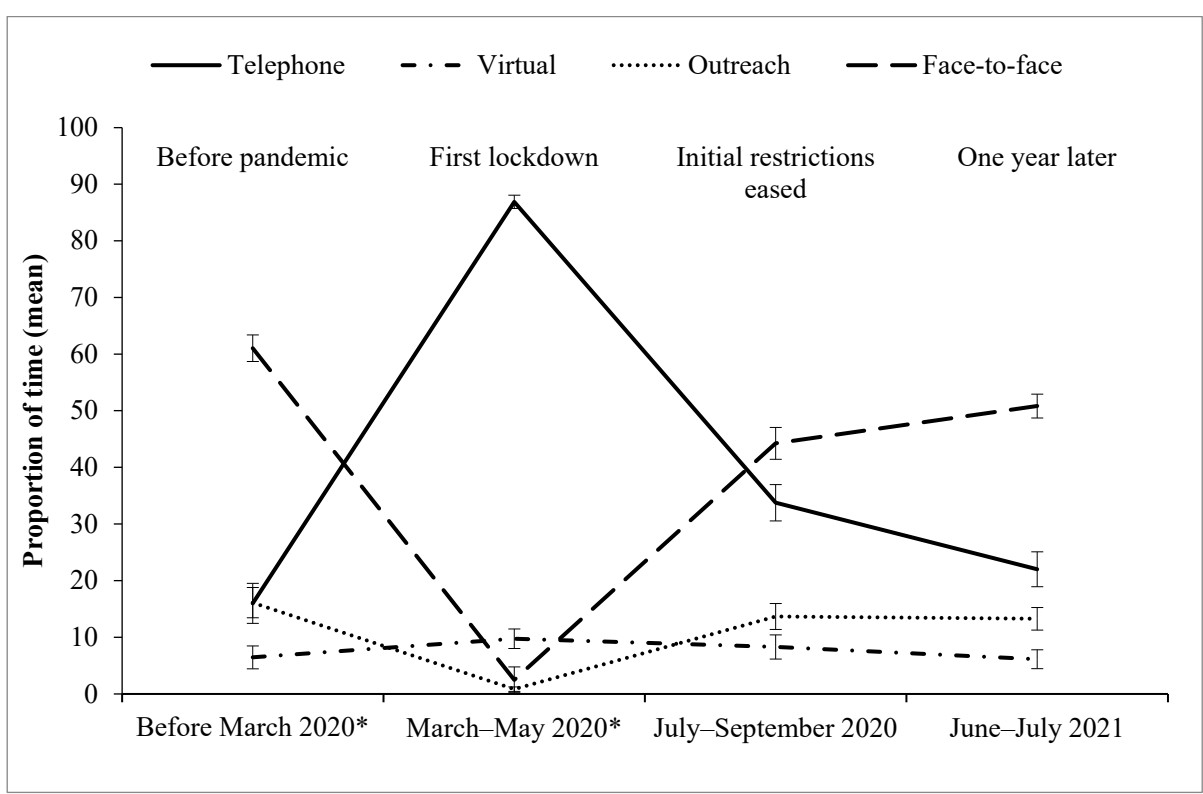

**Figure 1.** Staff-reported proportions of time spent providing counselling through different service modalities across the COVID-19 period. Responses for 'October 2019–February 2020' and 'March–May 2020' are retrospective estimates collected in July–September 2020 (denoted by *). Means and standard error bars are presented.

### 3.2. Service Treatment Record Data

The number of each counselling session by delivery type recorded in the service's electronic management system between October 2019 to July 2021 is shown in Figure 2. From October 2019 to February 2020, between 58–71% of recorded counselling sessions were delivered face-to-face. This decreased to 41% in March 2020 and to 5% by April 2020. Face-to-face counselling increased again to 30% by June 2020, and by August 2020 was of similar proportions to before lockdown (>58%). Proportions of outreach counselling were considerably lower, and remained consistently low throughout the entire recorded period (<17%). Telephone counselling was low, at <18%, until December 2019, then began to increase in January 2020 (36%) and peaked at 89% in April 2020. Following an inverse relationship to face-to-face counselling, telephone counselling considerably decreased by August 2020 and remained between 25–30% of recorded counselling sessions until July 2021. Virtual counselling remained at a negligible proportion of treatment sessions delivered (<3%). The total number of counselling episodes increased from March 2020 and remained at this higher level until October 2020.

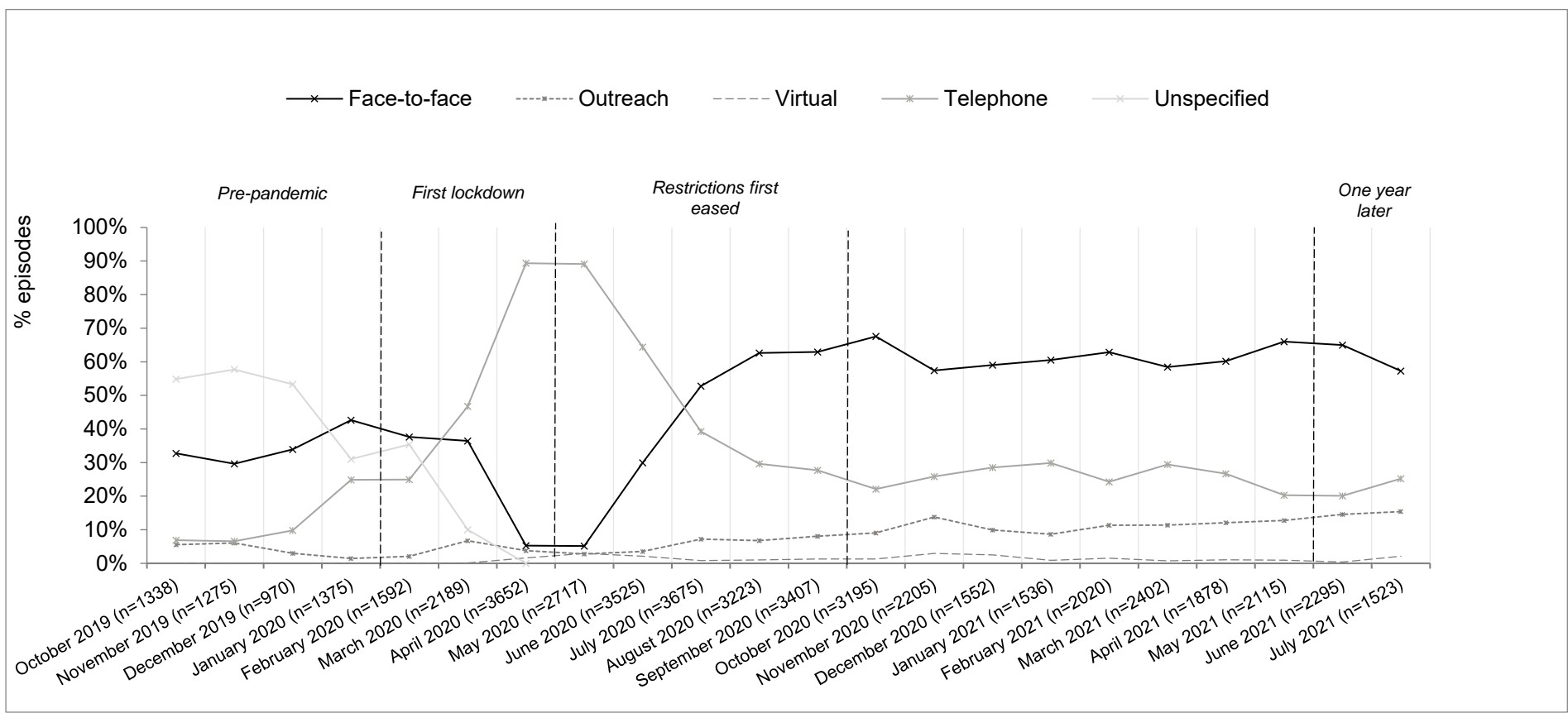

**Figure 2.** Service data on the percentage of monthly treatment sessions delivered face-to-face and by outreach, telephone, and virtual methods between October 2019–July 2021. Total episodes for each month are embedded within the x-axis. Data were collected and extracted by the service electronic management system. The computer system used by the service to record treatment sessions, including modality, was changed in April 2020. Prior to this change, specifying the treatment modality was not compulsory; hence the large number of 'unspecified' modalities from October 2019 to March 2020.

Mixed measure ANOVAs comparing treatment delivery type across the prepandemic period, the first lockdown, when restrictions were first eased, and one year later reported a significant main effect of delivery time and interaction with time (ANOVA results in Table S3). Bonferroni pairwise comparisons indicated an increase in the average number of face-to-face treatment delivery records during the first lockdown (March–May 2020) to when restrictions were first eased (July–September 2020) and one year later (June–July 2021) (pairwise comparisons table in Table S3). The average number of telephone treatment delivery records significantly increased from pre-pandemic (October 2019–February 2020) to the first lockdown, then significantly declined again when restrictions first eased and by one year later. The average number of outreach treatment delivery records was significantly higher from pre-pandemic to one year later. Virtual delivery did not change over the entire recording period.

## 4. Discussion

This study sought to examine the impact of the COVID-19 pandemic on the mode of AOD treatment delivery between October 2019 and July 2021. Both staff and electronic records indicated reductions in the delivery of face-to-face counselling during the first lockdown period in March–May 2020. This corresponded with increased use of telephone-delivered counselling, which declined after the initial lockdown period, yet remained higher than before the pandemic one year later. Virtually delivered counselling was negligible before the pandemic, and both staff and electronic records indicated no change. Although staff reported a small yet significant reduction in delivery of outreach counselling during the first lockdown, outreach remained low throughout the entire measurement period; however, electronic treatment records indicated a steady increase over time, which was significantly higher one year later. Together, these findings support increased provision of AOD counselling in response to the pandemic via telephone utilisation, but not virtual delivery options.

Telehealth approaches (telephone and virtual/video calls) minimised physical contact to prevent COVID-19 exposure for clients receiving treatment for AOD problems. With appropriate infrastructure and training [6], this method could be easily implemented for future restrictions or to protect vulnerable clients [7]. These options may have also reduced the anxiety associated with social contact among clients with physical and mental health vulnerabilities, who are at a greater risk of COVID-related harms [8]. While telephone-delivered counselling declined once restrictions were eased, the sustained use of this method since may indicate increased acceptability of telehealth among clients and clinicians. The extremely low use of virtual treatment in this study is inconsistent with other research (e.g., [11,12]), where virtual options were most prevalent and qualitatively suggested by clients as optimal [12].

Limitations of the study include the number of treatment staff that completed both annual surveys ($n$ = 117–221, 53%), which may reflect high staff turnover in the AOD treatment sector [13,14]. Furthermore, staff reports of treatment delivery pre-pandemic and during the initial lockdown were retrospective and are susceptible to recall bias; however, these reports matched the objective service-level electronic treatment record data. The recording of service information on treatment modality was not compulsory in the prepandemic phase of this study. Treatment was also specific to counselling and did not necessarily include higher-intensity treatments (e.g., cognitive behavioural therapy), where the use of video calls may be more prevalent.

To conclude, this AOD treatment service demonstrated changes to the delivery of care during and after the COVID-19 pandemic to allow for telehealth options. This supports the uptake of telehealth treatment-delivery training and infrastructure amid future virus-related outbreaks and associated restrictions. Future research should examine corresponding changes to client and staff preference for telehealth options, virtual/video options, or face-to-face treatment as a long-term consequence of the pandemic.

**Supplementary Materials:** The following supporting information can be downloaded at: https://www.mdpi.com/article/10.3390/biomed3020019/s1, Table S1: Significant changes in staff-reported delivery of different service modalities before, during, and after the onset of COVID-19 (M, SD); Table S2: Total count and percentages for treatment records collected between October 2019–July 2021. Table S3. ANOVA and pairwise comparison outcomes for electronic treatment records.

**Author Contributions:** The project was conceptualised by L.H., C.A.Q. and L.M.; M.C. led the writing of the manuscript and the analysis of the data, which was assisted by G.N. and L.M.; G.N. and L.M. helped to collect the data alongside R.E. and C.T. All authors have read and agreed to the published version of the manuscript.

**Funding:** The study was funded by the Australian Government Department of Health awarded to the National Centre for Youth Substance Use Research (NCYSUR) under the Drug and Alcohol Prevention Program. The funding source had no involvement in the study design; the collection, analysis, and interpretation of data; or the writing of the report.

**Institutional Review Board Statement:** The study was conducted in accordance with the Declaration of Helsinki, and approved by the participating AOD service and the University of Queensland Office of Research Ethics with Approval Codes: 2019002308 and 2020002322. Approval Dates: 17 October 2019 and 28 October 2020.

**Informed Consent Statement:** Informed consent was obtained from all subjects involved in the study.

**Data Availability Statement:** Data sharing will need to be requested and approved by the collaborating AOD service.

**Acknowledgments:** We would like to acknowledge Lives Lived Well, the collaborating treatment service, for facilitating the project.

**Conflicts of Interest:** The authors declare no conflict of interest. The funders had no role in the design of the study; in the collection, analyses, or interpretation of data; in the writing of the manuscript; or in the decision to publish the results.

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
