# Peer review of "A Longitudinal Assessment of Substance Use Treatment during the COVID-19 Pandemic Using Staff and Service Data"

_2673-8430, doi:10.3390/biomed3020019_

Round 1

Reviewer 1 Report

The importance of the topic is sound and the manuscript is well-written by and large. A few comments for improvement:

1. The title can be more concise. For instance, 'alcohol and other drug use' can be replaced with 'substance use'. Type of the study can be added as well. In addition, the abstract doesn`t report any perspective assessment and the title should be revised in this way as well.

2. Data collection methods should be described more in the abstract.

3. 'Discussion' should be replaced with 'Conclusion' in the abstract.

4. Place of the study should be added to the abstract.

5. Any abbreviation, like SM1, should be used in full-term for first time.

6. Ethical consideration and inclusion/ exclusion criteria should be added substantially.

7. Total number of participants in the abstract and results should be consistent.

Author Response

Reviewer 1

The importance of the topic is sound and the manuscript is well-written by and large. A few comments for improvement:

Thank you for taking the time to review our manuscript, and for your insightful recommendations that have improved the report.

  1. The title can be more concise. For instance, 'alcohol and other drug use' can be replaced with 'substance use'. Type of the study can be added as well. In addition, the abstract doesn`t report any perspective assessment and the title should be revised in this way as well.

We have now changed AOD to substance use in the title for conciseness, and adjusted the title in line with your recommendations to:

A longitudinal assessment of substance use treatment during the COVID-19 pandemic using staff and service data

  1. Data collection methods should be described more in the abstract.

We have now specified more detail on the collection methods in the abstract:

Two waves of online questionnaires were collected, with staff reporting on their time: before the pandemic (retrospectively for October 2019 - February 2020), during the first lockdown period (retrospectively for March-May 2020), when restrictions were initially eased (June - September 2020), and one year later (July 2021). Service records of the number of counselling episodes conducted by each treatment modality were extracted between October 2019 and July 2021, and analysed by month.

  1. 'Discussion' should be replaced with 'Conclusion' in the abstract.

Done.

  1. Place of the study should be added to the abstract.

We have now specified that the study took place in Queensland, Australia, in the abstract: “Treatment staff (n = 221) from a large AOD service in Queensland, Australia”

  1. Any abbreviation, like SM1, should be used in full-term for first time.

Done on page 2: “(staff demographics, Supplementary Material [SM] 1)”.

  1. Ethical consideration and inclusion/ exclusion criteria should be added substantially.

            We have now added detail regarding the inclusion/exclusion criteria for study

participation, as well as information on ethical considerations on page 2:

All clinical staff in a large substance use treatment service located in Queensland and New South Wales, Australia (approx. 204 services; 16,000 clients/year), were invited to complete an annual survey to obtain their views about relevant organisational issues. Inclusion criteria for participation in the current study was that staff had direct client contact as part of their role (either through provision or supervision of counselling). Eligible staff were electronically provided with information about the study, including that their participation was confidential and voluntary, and a consent form to indicate whether they provided consent to participate in the study or not. Information on staff consent and study participation remained with independent research staff only and was not accessible by members of the organisation.

Eligible and consenting staff who completed both survey occasions were included in this study (staff demographics, Supplementary Material [SM] 1). The two data collection points were July - September 2020 (n = 158), and June - July 2021 (n = 180). A total of 221 clinical staff completed the surveys; 117 (53%) completed both occasions, while 104 (47%) completed one occasion. Treatment record data for counselling episodes was extracted from the service’s electronic management system for the period of October 2019 – June 2021. Ethical approvals were obtained from the University of Queensland Office of Research Ethics (IDs 2019002308 and 2020002322)..”

  1. Total number of participants in the abstract and results should be consistent.

Thank you for this point. Following this and Reviewer 2’s similar comment, we have made efforts to increase the clarity of the sample size in the abstract, analysis section, and results. We now specify 117 members of staff in the abstract since this is the number of staff that were analysed, as follows: “Staff (n = 117) and service records indicated …”. We also specify this in the analysis (page 3):

Staff survey responses were analysed using a series of one-way, repeated measures ANOVAs (>2 response options) or t-tests (2 response options) for 117 staff that had completed the two survey waves. T-tests were also used to assess demographic differences between staff that had completed one (n = 104) compared with two (n = 117) surveys.”

In addition to the results section (page 3):

“While 117 (53%) of staff completed both waves of questionnaires, 104 (47%) staff only completed one of the two waves. When comparing differences between these staff, we did not find any significant differences in the age (t = 0.65, p = .514) or months of experience in the AOD sector”.

Reviewer 2 Report

Generally well written, however some details of the analysis and the depth of analysis could be improved. - ethics approval should state where ethical approval was obtained from rather than just "relevant human research ethics committee." So that the reader can access the approval. - the statistical analysis appears to have only analysed the 117 staff with both time points of the 221 total responses, however it is not explicitly stated at any time so is somewhat unclear. There is a section embedded in the staff demographics that examined the differences between the 117 staff who completed both surveys and the 103 that only completed one survey, however it needs to be a separate paragraph and explicitly state its purpose. - the service data is only described descriptively and there appears to be some scope for some statistical analysis of this data, especially as it is more complete than the survey data. I note that there is some reference to quality of treatment data pre-pandemic under limitations. However given the self-reported retrospective bias in survey data shouldn't limit the analysis of the treatment data for comparative purposes.

Author Response

Reviewer 2

Generally well written, however some details of the analysis and the depth of analysis could be improved.

Thank you for taking the time to review our manuscript, and for your insightful recommendations on how to improve it.

- ethics approval should state where ethical approval was obtained from rather than just "relevant human research ethics committee." So that the reader can access the approval.

This has been added to the following two locations in the manuscript:

Methods Page 2: “Ethical approvals were obtained from the University of Queensland Office of Research Ethics (IDs 2019002308 and 2020002322)”

Institutional Review Board Statement Page 8: “The study was conducted in accordance with the Declaration of Helsinki, and approved by the participating AOD service and the University of Queensland Office of Research Ethics with Approval Codes: 2019002308 and 2020002322. Approval Dates: 17/10/2019 28/09/2020. .

- the statistical analysis appears to have only analysed the 117 staff with both time points of the 221 total responses, however it is not explicitly stated at any time so is somewhat unclear. There is a section embedded in the staff demographics that examined the differences between the 117 staff who completed both surveys and the 103 that only completed one survey, however it needs to be a separate paragraph and explicitly state its purpose.

Thank you for this point. To increase clarity that 117 staff completed both time points, we have specified this in the abstract (“Staff (n = 117)”), and added this the methods section (page 2: “A total of 221 clinical staff completed the surveys; 117 (53%) completed both occasions, while 104 (47%) completed one occasion.”

Also in the results section we have now made this a separate paragraph and added specific details to increase clarity for readers:

“Page 3: While 117 (53%) staff completed both waves of questionnaires, 104 (47%) staff only completed one of the two waves. When comparing differences between these staff, there were no significant differences in the age (t = 0.65, p = .514) or months of experience in the AOD sector (t = 1.25, p = .214) between staff who completed one compared with two surveys. Staff who completed only one survey had been working in the service for significantly less months (M = 24.19, SD = 25.07) than staff who completed both surveys (t = 3.73, p<.001).”

We have also clarified this in the statistical analysis section:

Page 3: “Staff survey responses were analysed using a series of one-way, repeated measures ANOVAs (>2 response options) or t-tests (2 response options) for 117 staff that had completed the two survey waves. T-tests were also used to assess demographic differences between staff that had completed one (n = 104) compared with two (n = 117) surveys..”

- the service data is only described descriptively and there appears to be some scope for some statistical analysis of this data, especially as it is more complete than the survey data. I note that there is some reference to quality of treatment data pre-pandemic under limitations. However given the self-reported retrospective bias in survey data shouldn't limit the analysis of the treatment data for comparative purposes.

Thank you for this point, we have now included additional methods and analyses of the electronic treatment data to check staff responses corresponded with these. For a succinct analysis and for ease of comparison, we compared standardised count data for each treatment delivery type over the equivalent time periods reported by staff. The methods and results are now reported, as follows:

Methods – analysis page 3: Significant differences in treatment records were tested using mixed measure ANOVAs on the same four time periods as staff self-report data. Treatment episode counts for each month were stardardised to z-scores for each delivery type to account for fluctuation in the total number of records that was not attributable to the pandemic.

Results page 5: Mixed measure ANOVAs comparing treatment delivery type across pre-pandemic, first lockdown, when restrictions were first eased, and one year later reported a significant main effect of delivery time, and interaction with time (ANOVA results in SM3). Bonferroni pairwise comparisons indicated an increase in the average number of face-to-face treatment delivery records during the first lockdown (March – May 20) to when restrictions were first eased (July – September 2020) and one year later (June – July 2021) (pairwise comparisons table in SM3). The average number of telephone treatment delivery records significantly increased from pre-pandemic (Oct 2019 – Feb 2020) to the first lockdown, then significantly declined again when restrictions first eased and by one year later. Average number of outreach treatment delivery was significantly higher from pre-pandemic to one year later. Virtual delivery did not change over the entire recording period.

Some adaptations to the discussion have been made in line with these, however interpretations remain the same. Since there are several pairwise comparisons, these have been reported in a table contained in SM3.

Reviewer 3 Report

The manuscript is interesting .

However, the introduction needs reorganization and more details 

Author Response

Reviewer 3

The manuscript is interesting. However, the introduction needs reorganization and more details 

Thank you for taking the time to review our manuscript, and for your suggestions that have improved the report. We have made efforts to increase clarity in the introduction and have added additional details in the literature review and description of the study. We have also added additional literature (citations 5 and 9) that do not come up as tracked changes when using the reference manager.

Reviewer 4 Report

Well-constructed study and well-illustrated results. Nothing to report

Author Response

Reviewer 4

Well-constructed study and well-illustrated results. Nothing to report

Thank you for taking the time to review our manuscript, and that you are pleased with the document.
